# Dendritic Cell Vaccines Impact the Type 2 Innate Lymphoid Cell Population and Their Cytokine Generation in Mice

**DOI:** 10.3390/vaccines11101559

**Published:** 2023-10-03

**Authors:** Lily Chan, Yeganeh Mehrani, Jessica A. Minott, Byram W. Bridle, Khalil Karimi

**Affiliations:** 1Department of Pathobiology, Ontario Veterinary College, University of Guelph, Guelph, ON N1G 2W1, Canada; lchan12@uoguelph.ca (L.C.); ymehrani@uoguelph.ca (Y.M.); minott@uoguelph.ca (J.A.M.); 2Department of Clinical Science, School of Veterinary Medicine, Ferdowsi University of Mashhad, Mashhad P.O. Box 91775-1111, Iran

**Keywords:** type 2 innate lymphoid cell, dendritic cell, cytokine, communication, dendritic cell vaccine

## Abstract

Dendritic cell (DC) vaccines can stimulate the immune system to target cancer antigens, making them a promising therapy in immunotherapy. Clinical trials have shown limited effectiveness of DC vaccines, highlighting the need to enhance the immune responses they generate. Innate lymphoid cells (ILCs) are a diverse group of innate leukocytes that produce various cytokines and regulate the immune system. These cells have the potential to improve immunotherapies. There is not much research on how group 2 ILCs (ILC2s) communicate with DC vaccines. Therefore, examining the roles of DC vaccination in immune responses is crucial. Our research analyzed the effects of DC vaccination on the ILC2 populations and their cytokine production. By exploring the relationship between ILC2s and DCs, we aimed to understand how this could affect DC-based immunotherapies. The results showed an increase in the number of ILC2s in the local draining lymph node and spleen of tumor-free mice, as well as in the lungs of mice challenged with tumors in a pulmonary metastasis model. This suggests a complex interplay between DC-based vaccines and ILC2s, which is further influenced by the presence of tumors.

## 1. Introduction

Innate lymphoid cells (ILCs) are a type of leukocyte that can be divided into three groups: ILC1, ILC2, and ILC3. They are considered to be the innate equivalent of T helper (Th) cells, with each subset sharing similar transcription factors, phenotypic markers, and cytokine profiles as Th1, Th2, and Th17, respectively. ILCs are different from Th cells in that they lack antigen receptor specificity [1]. Research has shown that ILCs have a complex biology and can exhibit plasticity between subsets [2]. Despite their potential in immunotherapies, their full capabilities are yet to be harnessed, and more research is necessary to determine how to manipulate their roles in immune responses effectively. Understanding the communication and functions of ILCs in health and disease is essential to achieve this goal. Environmental cues, such as cytokine milieus, greatly influence the phenotype and function of ILCs, and their contributions to immune responses are highly context-dependent, as explained previously [3].

Dendritic cell (DC)-based vaccines are an immunotherapy platform that utilizes the ability of DCs to activate, educate, and fine-tune innate and adaptive immunity. DCs are potent antigen-presenting cells, and they play a crucial role in educating antigen-specific cytotoxic T cells. This is the foundation of the DC vaccines [4]. Priming tumor-specific responses via the effector cells of the adaptive immune system is a key goal of immunotherapy. Hence, many immunotherapies aim to enhance adaptive immunity, including chimeric antigen receptor-T cell therapy and immune checkpoint blockade therapies that mainly target T cell checkpoints. However, in designing immunotherapies, the significance of the contributions of innate components of the immune system has become more of a consideration [5]. Natural killer (NK) cells have gained significant attention, particularly in DC immunotherapies [6,7,8]. Since NK cells are a subset of ILCs and ILCs have many similarities to T cells, exploring the potential of other ILC subsets in immunotherapies has become an active area of research [1,9].

ILC2s have important roles in immunity against extracellular parasites and inflammation [10]. When stimulated via thymic stromal lymphopoietin, interleukin (IL)-25, and IL-33, they produce various cytokines such as IL-4, IL-5, IL-9, and IL-13 [11]. Although ILC2s are typically linked to protecting the mucosa, specifically in anti-helminthic immunity [12], in certain cancers, they have been shown to support myeloid-derived suppressor cells [13,14] and impair NK cell activity [15,16], which contribute to immunosuppression, leading to tumor progression. ILC2s were detected at relatively high frequencies in breast [17], bladder [14], and gastric [18] cancers. Conversely, Huang et al. [19] observed that the increased infiltration of ILC2s into tumors correlated with smaller tumors in mice. Furthermore, they observed that gene signatures in ILC2s were associated with better survival in human patients with colorectal cancers [19]. Additionally, Saranchova et al. [20] noted that ILC2s in tumor microenvironments participated in immunosurveillance and potentially supported immune responses that could prevent metastases and limit the growth of tumors. Also, recombinase-activating gene (RAG)-knockout mice showed increased tumor growth compared to RAG-knockout mice treated with adoptively transferred ILC2s, suggesting that ILC2s support anti-tumor responses. The mice into which ILC2s were transferred showed an increase in eosinophil infiltration and collagen deposition within their tumors. [21]. Therefore, ILC2s can promote both pro- and anti-tumor responses, indicating their potential for manipulation via immunotherapy to achieve the desired outcomes due to their plasticity.

The current research on the communication between endogenous DCs and ILC2s is insufficient, and the interaction between the DC employed as vaccines and host ILC2s remains unclear. Nonetheless, studies reveal that ILC2s in the lungs trigger host DC migration to the local draining lymph nodes by producing IL-13. These recruited DCs by ILC2s are crucial in promoting adaptive immune responses [22,23,24]. The findings indicate that ILC2s can initiate immune responses and suggest the existence of important interactions between host DCs and ILC2s. The discovery opens up possibilities for exploring the potential communication between DCs used as a vaccine and host ILCs, which could trigger additional immune response loops involving host DCs [25].

DC cancer vaccines have generated significant interest in the field of immunotherapy since the first vaccine of this type received approval from the US Food and Drug Administration in 2010. This vaccine has been used to combat prostate cancers with promising results. [26]. However, the effectiveness of DC vaccination has been limited, which means that there is a need to improve the immune response against tumors elicited by DC vaccines. To achieve this, it is desirable to enhance beneficial cell-to-cell communications in the host to modulate the DC vaccine platform. Interactions between DC vaccines and host leukocytes need to be well defined. Further research is required to investigate the communication between DCs and ILC2s. It is currently unknown how optimizing the design of DC-based immunotherapies may impact ILC2s. This study aimed to determine if DC vaccination affects ILC2s in naïve hosts and during a tumor challenge. This is an essential initial step in uncovering the potential communication between these innate leukocytes.

## 2. Materials and Methods

### 2.1. Mice

We received a group of female C57BL/6 mice from Charles River Laboratory when they were 36 to 52 days old. They were kept in the animal isolation unit at the University of Guelph, where the environment was controlled. Before starting the experiments, the mice had at least one week to acclimatize to the isolation facility. They were given unlimited access to food and water.

### 2.2. Dendritic Cell Cultures

We collected bone marrow from naïve female C57BL/6 mice. We harvested their femurs and tibias, removed the ends of the bones, and flushed the bone marrow out with a syringe and phosphate-buffered saline. We then made the bone marrow into a single-cell suspension by pipetting it, counted the cells using a hemocytometer, and resuspended them in media (RPMI with 1% penicillin/streptomycin, beta-mercaptoethanol, and 10% fetal bovine serum) at a concentration of 1.25 × 10^6^ cells per mL. We added the granulocyte–monocyte-colony-stimulating factor (GM-CSF) (Biolegend Cat#576308) at a concentration of 20 ng/mL. We aliquoted the cultures into flasks, which we kept in a 37 °C humidified incubator with 5% CO_2_ starting from day zero. On day two, we added fresh media and GM-CSF to the cultures. On day five, we removed half of the media and centrifuged it at 500× *g* for 5 min to recover non-adherent cells. We discarded the supernatant and resuspended the cells in the same volume of fresh media and GM-CSF that we had removed. On day seven, we used the cultures for vaccine preparation.

### 2.3. Dendritic Cell Vaccine Preparation

To harvest the DC cultures, non-adherent and loosely adherent cells were pipetted up and down using a serological pipette. The cells were then transferred into a conical tube and centrifuged at 500× *g* for 5 min. The cells were resuspended in 5 mL of media and counted using a hemocytometer. The DCs were then stimulated with 100 ng/mL of lipopolysaccharide (LPS) from Escherichia coli O55:B5 and loaded with 1 µg/mL of the chicken ovalbumin_257–264_ (SIINFEKL) peptide (PepScan Systems, Lelystad, Netherlands), which is the immunodominant CD8^+^ T cell epitope for C57BL/6 mice. This process was carried out for one hour at 37 °C and mixed every 10–15 min. The cells were then washed three times with phosphate-buffered saline and resuspended in phosphate-buffered saline for 1 × 10^6^ cells per 60 µL. The mice were then injected with 0.5 × 10^6^ cells per injection dose into the footpad of a hind limb.

### 2.4. Flow Cytometry Gating Strategy

To identify the ILC2s, we first used the forward scatter area (FSC-A) and the side scatter area (SSC-A) to gate lymphocytes. Then, we eliminated doublets by comparing the FSC-height (FSC-H) versus the FSC-A. We removed the dead cells by using a fixable viability dye. Next, we defined ILC2s as CD45^+^ Lineage- CD90.2^+^ ICOS^+^ ST2^+^ cells. To exclude major hematopoietic cell lineages like T and B lymphocytes, monocytes/macrophages, granulocytes, NK cells, and erythrocytes, we used the mouse hematopoietic lineage cocktail (Lineage) and cell-surface antigens. Appendix A shows representative dot plots of our gating strategies.

### 2.5. Intracellular Cytokine Antibody Staining

The cells were fixed and incubated at 4 °C for 20 min. After fixation, they were washed twice with a permeabilization buffer (BioLegend Cat#421002, San Diego, CA, USA). Next, the cells were resuspended in a master mix of intracellular-staining antibodies (anti-IL-13, eBioscience Cat#12-7133-82 PE, San Diego, CA, USA; anti-IL-5, BioLegend Cat#504305 APC) and incubated at 4 °C for 20 min. The cells were then washed twice with the permeabilization buffer, resuspended in a FACs buffer, and analyzed using a BD FACS Canto II flow cytometer and the FlowJo^TM^ v9 software.

### 2.6. Tissue Processing

We collected and prepared spleens, lungs, and popliteal lymph nodes by placing them in Petri dishes with Hank’s balanced salt solution. Using a syringe stopper, we pressed the spleens and lymph nodes into a single-cell suspension and filtered it through a 70 μm cell strainer to remove debris. To remove the red blood cells from the spleens, we used a ACK lysing buffer (8.29 g NH4Cl [0.15 M], 1 g of KHCO3 [10.0 mM], 37.2 mg of Na2EDTA [0.1 mM] in 1 mL of Milli Q water), then washed the samples with Hanks balanced salt solution before resuspending them in phosphate-buffered saline. We counted the cells and seeded them into 96-well round-bottom plates. We also perfused the lungs with 5 mL of phosphate-buffered saline during harvesting to remove the blood, weighed them, and enzymatically digested them with collagenase IV (1 mg/mL) and DNase I (5 μg/mL) in Hanks balanced salt solution. After incubating the processed lungs at 37 °C for 20 min, we filtered the cell suspension through a 70 nm cell strainer into a 50 mL conical tube and removed any remaining red blood cells with the ACK lysis buffer. Finally, we washed the cells twice with Hanks balanced salt solution and seeded them into a 96-well round-bottom plate.

### 2.7. DC Vaccine Migration to Lymph Node

The DC vaccines were prepared as described in Section 2.3 and then resuspended in phosphate-buffered saline to a concentration of 1 × 10^6^ cells per mL. To label the cells, Carboxyfluorescein succinimidyl ester (CFSE) was added at a concentration of 10 μM, mixed thoroughly, and incubated for 15 min in a water bath set at 37 °C. To dilute and wash the cells, ten times the volume of phosphate-buffered saline was used. The cells were then washed with 10 mL of media once, and with phosphate-buffered saline twice. The resulting cell concentration of 1 × 10^6^ cells per 30 μL of phosphate-buffered saline was injected into the footpads of mice, with 30 μL per footpad. After 24 h, the popliteal lymph nodes were harvested, processed, and plated in 96-well plates. The cells were then resuspended in Fc block (anti-CD16/CD32, BioLegend Cat#101320) and incubated for 15 min at 4 °C. The cells were washed twice with the FACS buffer and resuspended in a mastermix of surface-stain antibodies (CD11c). Finally, the DCs that had migrated (CFSE + CD11c + cells) were quantified using FACS analysis.

### 2.8. Cytokine Response Assay

To prepare for flow cytometry analysis, single-cell suspensions from the spleens and lungs were stimulated with phorbol myristate acetate (PMA) at 10 ng/mL and ionomycin at 1500 ng/mL. The cells were then placed in an incubator at 37 °C and Brefeldin A (GolgiPlug, Biolegend Cat#420601) was added at a 1000× dilution after one hour. Following an additional four hours, the cells were washed with phosphate-buffered saline and stained for analysis using flow cytometry.

### 2.9. B16F10 Cell Culture, Tumor Challenge, and Tumor Nodule Quantification

To prepare the B16F10 cells, they were taken out from liquid nitrogen, counted using a hemocytometer, and then resuspended in 10 mL of media (Dulbecco’s Modified Eagle Medium with 10% bovine calf serum and 1% penicillin/streptomycin) at a concentration of 1 × 10^6^ cells. The cells were then divided into culture flasks and placed in a 37 °C humidified incubator with 5% CO_2_. Later, the cells were washed and resuspended in phosphate-buffered saline at a concentration of 3 × 10^5^ cells per 200 µL. To assess ILC2s, a dose of B16F10 was administered intravenously 7 days after DC immunization. The mice were then sacrificed 3 days after the challenge with 1 × 10^6^ B16F10 melanoma cells. For Appendix A, 1 × 10^6^ B16F10 melanoma cells were injected intravenously 7 days after immunization, and the mice were sacrificed after 14 days. The lungs of the mice were removed, fixed in 2% paraformaldehyde, and kept in 70% ethanol before being counted under a dissecting microscope on day 14 after the tumor challenge to enumerate the lung metastatic nodules.

### 2.10. Statistical Analysis

The data analysis was conducted using GraphPad Prism version 9. If the *p*-values were less than 0.05, it was determined that the means for the treatment groups were significantly different. In the experiments with two treatment groups, the Student’s unpaired two-tailed *t*-test was utilized. For the experiments that had multiple iterations of a single variable, a two-way analysis of variance and Šídák’s multiple comparisons test were employed.

## 3. Results

### 3.1. DC-Based Vaccines Efficiently Induced an Increase in the Number of ILC2s in Draining Lymph Nodes and Spleen

Upon migration to the local draining lymph nodes, the DCs induce cellular changes, including increasing the number of local NK cells and T cells [27,28]. This concept is upheld in DC vaccination, where the vaccine is designed to induce anti-tumor immune responses [29]. We demonstrated that in the following inoculation of 1 × 10^6^ of bone marrow-derived DCs carrying the ovalbumin_257–264_ peptide into the footpads of mice, there was an increase in the total number of cells in the lymph node (Figure 1a) supporting the concept that there are cellular changes occurring in the local draining lymph node following the DC vaccination [6]. Furthermore, increases in the total number (Figure 1b) and percentage (Figure 1c) of ILC2s in the lymph node were observed from days two to seven post-immunization.

Since a local change in the ILC2 populations was observed, the potential for systemic changes in the ILC2 populations was investigated. The spleen is a specialized immunological organ that plays a significant role in the local and systemic innate and adaptive immunity [30]. The spleen is the largest secondary lymphoid organ [31] and is distal from the site of injection of the vaccine, which was injected into the footpad of a hind limb. Following the DC vaccination, there was an increase in the total number (Figure 1e) and percentage (Figure 1f) of ILC2s in the spleen. The spleen was also examined for the production of IL-13 and IL-5 by ILC2s using an ex vivo stimulation method with phorbol myristate acetate (PMA) and ionomycin. The DC-vaccinated mice had an increase in the number of ILC2s producing IL-13 and IL-5 (Figure 1h–j). This indicates that DC vaccination influenced both the quantity and quality of ILC2s in the spleen and extended our findings to suggest that both local systemic changes in ILC2s occur.

### 3.2. The DC Vaccine Elicited Cytokine Secretion via Splenic ILC2s after Mice Were Challenged Intravenously with B16F10 Melanoma Cells

Their environment and external stimuli highly influence ILC2s. Therefore, the influence of DC vaccination on the ILC2 populations was investigated in a B16F10 murine melanoma challenge model to determine if the communications observed in naïve mice apply in a cancer-bearing host. Since adaptive immunity and antigen education were not being investigated in this model, a tumor cell line that was irrelevant to the DC vaccine antigen was chosen. This also limited the influence of adaptive immunity in the experimental evaluations. There was no significant increase in the number or proportion of splenic ILC2s after the intravenous administration of B16F10 cells (Figure 2).

We demonstrated that there is no increase in the total number or percentage of ILC2s in the spleen. However, an increase in ILC2s secreting IL-13 and IL-5 in the spleen was detected (Figure 2). Therefore, this indicates that the DC vaccination participated in priming the ILC2s in the spleen to have a more robust cytokine response.

### 3.3. The Number and Proportion of Cytokine-Producing Pulmonary ILC2s Increased after DC Vaccination Followed by Intravenous Challenge with B16F10 Melanoma Cells

The lungs are a favored site of metastasis for B16F10 cells when administered intravenously in mice [32]. Therefore, the lungs of DC-vaccinated mice were also examined for ILC2s following the tumor challenge. There was an increase in the total number and percentage of ILC2s in the lungs following the challenge with B16F10 cells. Furthermore, the number of IL-5- and IL-13-producing ILC2s in the lungs of the DC-vaccinated mice significantly increased (Figure 3) compared to the mice that did not receive a DC vaccine. This indicates that the DC vaccination influenced ILC2s in the lungs, both in terms of their numbers and their functional cytokine profiles. This further suggests that DC vaccines communicate with ILC2s and can influence their responses.

## 4. Discussion

The immunotherapies were based on the DC platform function via the modulation of the immune system. This includes licensing the innate effector mechanisms and inducing adaptive antigen-specific immune responses. We and others have now reported cellular changes in the local draining lymph nodes after the administration of DC vaccines. We showed that the inoculation of DCs into the footpads of mice led to the migration of the injected DCs into regional lymph nodes, where they modified the overall cellularity (unpublished data; Appendix A). We have also shown that the number of NK cells in the draining lymph nodes increased after immunization with DCs [6,33]. In this study, our aim was to determine if DC vaccines influence ILC2 responses. If they do, this could represent a novel avenue for modulating immune responses to optimize immunotherapies. The recruitment of DCs by ILC2s plays a crucial role in promoting adaptive immune responses [22,23,24], offering a promising approach to optimize immunotherapies by modulating immune responses. This potential communication between the DCs used as a vaccine and host ILCs, recruited after vaccine administration, may trigger additional immune response loops involving host DCs. We documented an increase in the number and proportion of ILC2s both locally in the draining lymph nodes as well as systemically, as assessed in the spleen (Figure 1), following the administration of a DC vaccine. This suggested that the DC vaccine influenced ILC2s via migratory and/or recruitment mechanisms or by stimulating the local proliferation of ILC2s. These results suggest that direct or indirect communications occur between DCs and ILC2s in tumor-free mice. The active communication between DC vaccines and ILC2s were further supported by increased splenic ILC2s (Figure 1), especially those secreting IL-5 and/or IL-13 (Figure 1) one week after immunizing the mice with DCs.

We conducted additional research on the relationship between DCs and ILC2s using a mouse model of metastatic lung cancer. Our previous studies have demonstrated that vaccination with DCs can trigger a quick expansion of NK cells capable of releasing interferon-γ. This, in turn, mediated a reduction in tumor nodules in a murine melanoma pulmonary metastatic model [6]. Using the same cancer model, our results showed that host DCs were required for protective immunity against the tumor challenge that was mediated by NK cells [33]. We noticed that DC vaccines offer complete protection against tumors, whether loaded with antigens or given as empty DCs [6,33] (Appendix A). Based on this, we used the model not to assess the effect of ILC2s on the efficacy of DC vaccination in the pulmonary metastatic model of murine melanoma, but to dissect the basic aspects of the responses of ILC2s during the tumor challenge (Figure 2 and Figure 3). Additional tumor models would be necessary to study the effects that ILC2s could have on the tumor burden following the immunization with DCs. After being vaccinated, the mice were given an intravenous challenge with B16F10 melanoma cells. Three days later, their spleens and lungs were checked for ILC2 responses. The tissues were of interest because the spleens represent the largest secondary lymphatic organ, and the lungs are a preferential location of metastasis of this tumor cell line [32]. The higher number of splenic ILC2s in DC-vaccinated mice, compared to those that were not immunized that had previously been observed on day seven in tumor-free mice (Figure 1), was not detectable on day 10 in mice that had received the tumor challenge (Figure 2). This might indicate that splenic ILC2s migrated to other locations, such as developing tumor microenvironments seeded by B16F10 cells to contribute to immune responses. Another possibility is that the effect of DC immunization on the ILC2 populations was only transient and ILC2-mediated responses were no longer detectable ten days post-vaccination. The former mechanism is supported by the fact that ILC2s were more numerous in the lungs of mice that had been challenged with melanoma cells (Figure 3). Another potential explanation for an undetected increase in ILC2s three days after the tumor challenge could be because the growth of ILC2s was hindered by the challenge or the transformation of the ILC2s into other ILC2 subsets in the spleen. This could be due to alterations in the microenvironment that affect ILC plasticity. Tissue-resident ILCs sense and adapt to environmental changes, and ILCs can adopt new phenotypic and functional profiles [3,34].

The enhanced production of IL-5 and IL-13 was present in the splenic ILC2s of the tumor-challenged mice regardless of the disappearance of increased numbers of splenic ILC2s over time, which had been observed in tumor-free mice upon the tumor challenge (Figure 2). This indicates that the DC vaccine had a lasting impact on the cytokine production ability of splenic ILC2s, irrespective of their overall numbers. This effect lasted for at least 10 days and was not affected by tumor-induced suppression. Furthermore, DC vaccination also affected pulmonary ILC2s ten days after immunization with DCs and three days after the challenge with B16F10 melanoma cells. In the lungs of vaccinated animals, ILC2s were more numerous, including IL-5- and/or IL-13-producing ILC2s (Figure 3). This indicated that the DC vaccine systemically enhanced cytokine production by ILC2s 10 days following the DC immunization, followed by three days after the intravenous melanoma challenge.

We previously published research in a pulmonary metastatic murine melanoma model in which a DC vaccine reduced the number of lung nodules in both prophylactic and therapeutic scenarios [6]. This benefit was dependent on NK cells and NK/CD8^+^ T cells, respectively. Presently, DC-based vaccines are administered to patients that already have cancer. Despite this, DC vaccines have been most studied in prophylactic settings [35,36,37]. However, there may be applications where prophylactic immunization with DCs might be beneficial (e.g., in people genetically predisposed to cancer due to BRCA mutations) and are a legitimate avenue of investigation for DC immunotherapies.

More functional studies are needed to investigate the DC vaccine-mediated induction of ILC2-produced cytokines IL-5 and IL-13 in both prophylactic and therapeutic scenarios. But, this would require something other than blocking antibodies since the systemic blockade of IL-5 and IL-13 would definitively not prove that ILC2s were the exclusive source of the cytokines. Furthermore, functional studies of ILCs would require genetically modified mice and the use of adoptive cell transfer to facilitate the in vivo isolation or depletion of ILCs. The cytokine-producing similarities of ILCs and Th cells, combined with the plasticity of both cell types, would require intensive experiments to parse out their respective contributions. This would include fate mapping, the use of genetically engineered mice, and adoptive cell transfer, each of which has important limitations when studying ILCs because of their elusive nature. Our current study has established a biological relationship between DCs and ILC2s that provides a strong rationale for the design of future functional studies.

The relevance of intratumoral ILC2s has been only partially verified. This is, at least in part, attributable to the limited amount of research that has been conducted on ILC2s and cancers, in addition to the contrasts in the findings of the research that has been performed to date. Even though there has been little research into how ILC2s influence tumorigenesis and/or progression, ILC2s and/or their signature cytokines, such as IL-5 and IL-13, have been shown to be involved in pro- [13,14,15,16,18,38,39,40] and anti-tumor [19,20,21] immune responses. Since ILC2s have a plasticity that can be influenced by external stimuli, and they can promote pro- and anti-tumor effects, there is potential to optimize their phenotype during immunotherapies to bias them towards phenotypes that can promote tumor regression. The influence of ILC plasticity would also need to be studied and examined further for how it could contribute to the results observed. Extensive experiments and studies would be needed to study ILC plasticity in these contexts and expand upon the findings of this paper, including fate mapping, RNA sequencing, and the evaluation of other leukocytes and the ILC populations.

## 5. Conclusions

In this study, we showed that vaccination using DCs can affect ILC2 responses both locally and systemically. However, it is currently unknown whether these responses are helpful or harmful in the context of cancers. This is the direction that future research in this area will take. To the best of our knowledge, there has been no prior research focusing on ILC2s during immunization with DCs. Additional research into whether the ILC2 responses support pro- or anti-tumor responses could help inform the methods of optimizing communications between DC vaccines and ILC2s as a way to improve cancer immunotherapies.

## Figures and Tables

**Figure 1 vaccines-11-01559-f001:**
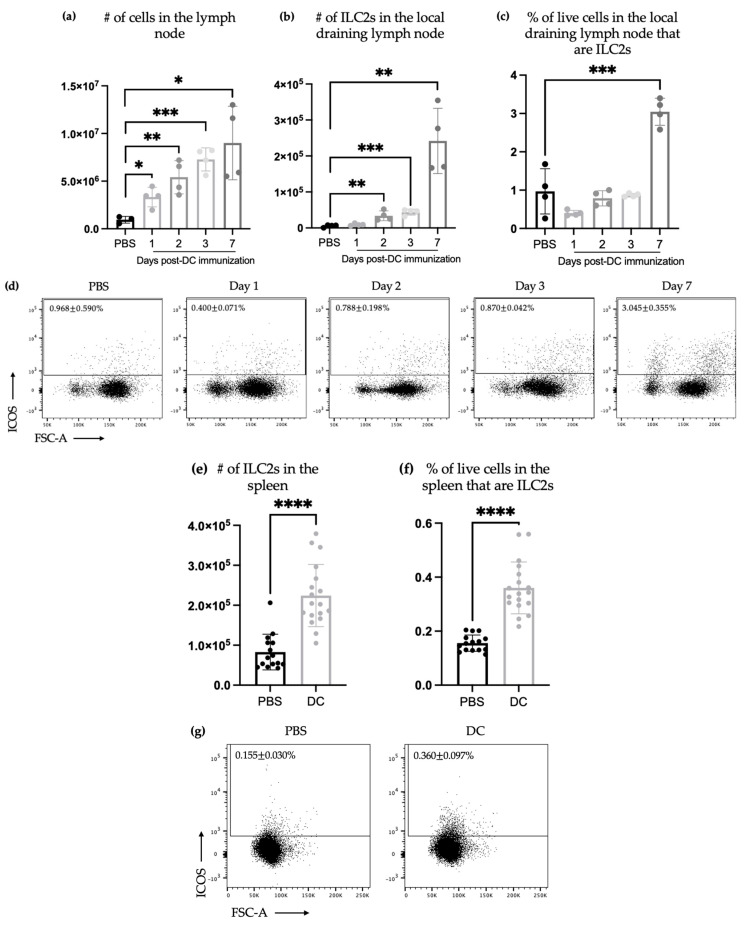
The number and proportion of ILC2s in local draining lymph nodes increase after DC immunization. Two mice in each group of female C57BL/6 mice were given DC vaccines via injection into a hind footpad. Popliteal lymph nodes were collected and examined for ILC2s’ accumulation on the specified days after injection. (**a**) The number of cells from popliteal lymph nodes was counted. (**b**) Accumulation of CD45^+^Lineage^-^CD90.2^+^ST2^+^ICOS^+^ ILC2s in the local draining lymph nodes was monitored via flow cytometry (Appendix A) and the (**c**) percentage of live cells in the lymph node that were ILC2s were determined. Each bar represents the data from the popliteal lymph nodes of two mice. At each time point post-immunization, a one-way analysis of variance was used to determine significant differences relative to the control group that was inoculated with phosphate-buffered saline (* *p*-value < 0.05, ** *p*-value < 0.005, and *** *p*-value < 0.0005). (**d**) The flow cytometry dot plots display the percentage of ILC2s present. DC immunization resulted in an increase in both the number and proportion of ILC2s, as well as the number of cytokines producing ILC2s within the spleen. Female C57BL/6 mice received DC vaccines via injection in the footpad of a hind limb. After one week, spleens were collected and analyzed for the presence of ILC2 accumulation. The quantity of splenic ILC2s was gauged based on (**e**) the total number and (**f**) percentage of overall cells from 18 DC-immunized mice and 15 control mice from four experimental replicates that had been given phosphate-buffered saline inoculations (**** *p*-value < 0.0001). The data were analyzed using Student’s t-test. Accumulation of CD45^+^Lineage^-^CD90.2^+^ST2^+^ICOS^+^ ILC2s in the spleen was monitored via flow cytometry analysis (Appendix A). (**g**) Representative flow cytometry dot plots showing the percentage of ILC2s. The spleens were also analyzed via intracellular cytokine staining for ILC2s producing IL-5 and IL-13. Shown are the number of splenic ILC2s that were producing (**h**) IL-13, (**i**) IL-5, or (**j**) IL-5 and IL-13. The data were analyzed via a two-way analysis of variance; *** *p*-value < 0.005; and **** *p*-value < 0.0001. (**k**) Accumulation of CD45^+^Lineage^-^CD90.2^+^ST2^+^ICOS^+^ ILC2s producing cytokines in the spleen was monitored via flow cytometry. Representative dot plots showing the percentage of ILC2s that were IL-5^+^ and/or IL-13^+^ are shown. The error bars displayed in the graphs represent the standard deviation.

**Figure 2 vaccines-11-01559-f002:**
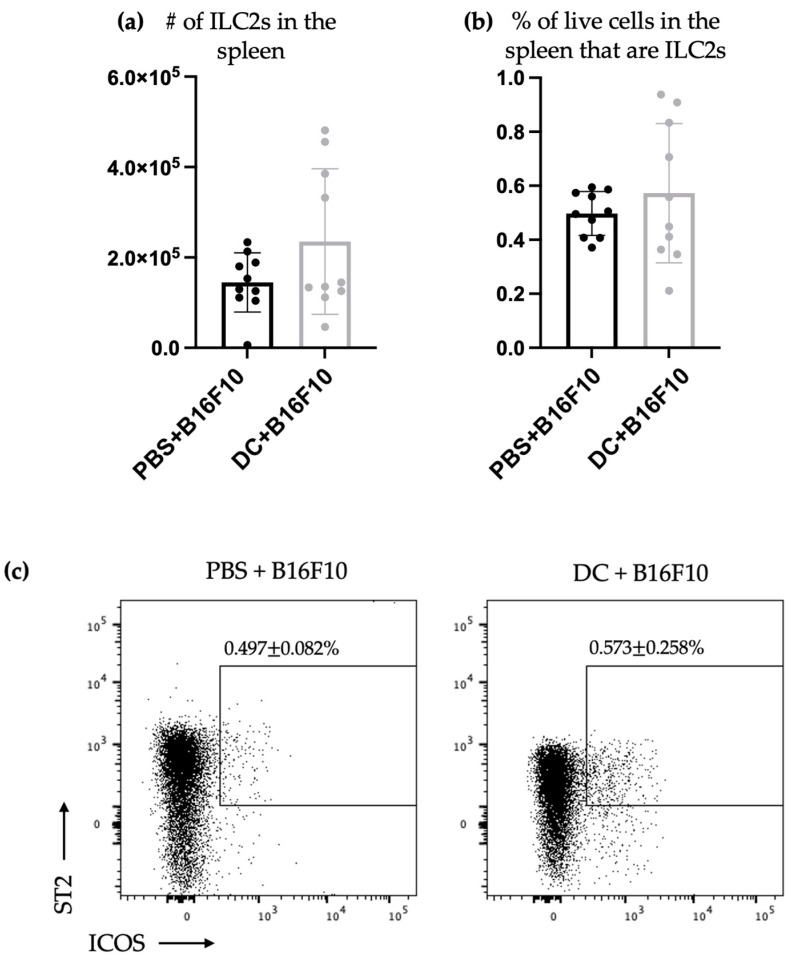
Neither the total number nor percentage of splenic ILC2s changed in DC-immunized mice after challenge with B16F10 melanoma cells, but there was an increase in cytokine secretion by the splenic ILC2s. Female C57BL/6 mice were inoculated with DC vaccines via injection into a hind footpad, and one-week later, B16F10 cells were injected into the tail vein. Spleens were harvested three days after challenge and analyzed for accumulation of ILC2s. The (**a**) number and (**b**) proportion of splenic ILC2s were enumerated from DC-immunized mice and control mice treated with phosphate-buffered saline (*n* = 10/group). The data were analyzed via Student’s t-test. Means were not significantly different. (**c**) Accumulation of CD45^+^Lineage^-^CD90.2^+^ST2^+^ICOS^+^ ILC2s in the spleen was monitored via flow cytometry (Appendix A). Representative dot plots show the percentage of ILC2s. Spleens were also analyzed for potential cytokine-producing ILC2s via ex vivo stimulation with PMA and ionomycin and flow cytometry after intracellular cytokine staining. Shown are the number of ILC2s producing (**d**) IL-13, (**e**) IL-5, (**f**) IL-13 and IL-5. The data were analyzed using A two-way ANOVA test ** *p*-value < 0.005, *** *p*-value < 0.0005, and **** *p*-value < 0.0001. (**g**) Accumulation of CD45^+^Lineage^-^CD90.2^+^ST2^+^ICOS^+^ ILC2s producing cytokines in the spleen was monitored via flow cytometry. Representative dot plots show the percentage of ILC2s that were IL-5^+^ and/or IL-13^+^. The error bars displayed in the graphs represent the standard deviation.

**Figure 3 vaccines-11-01559-f003:**
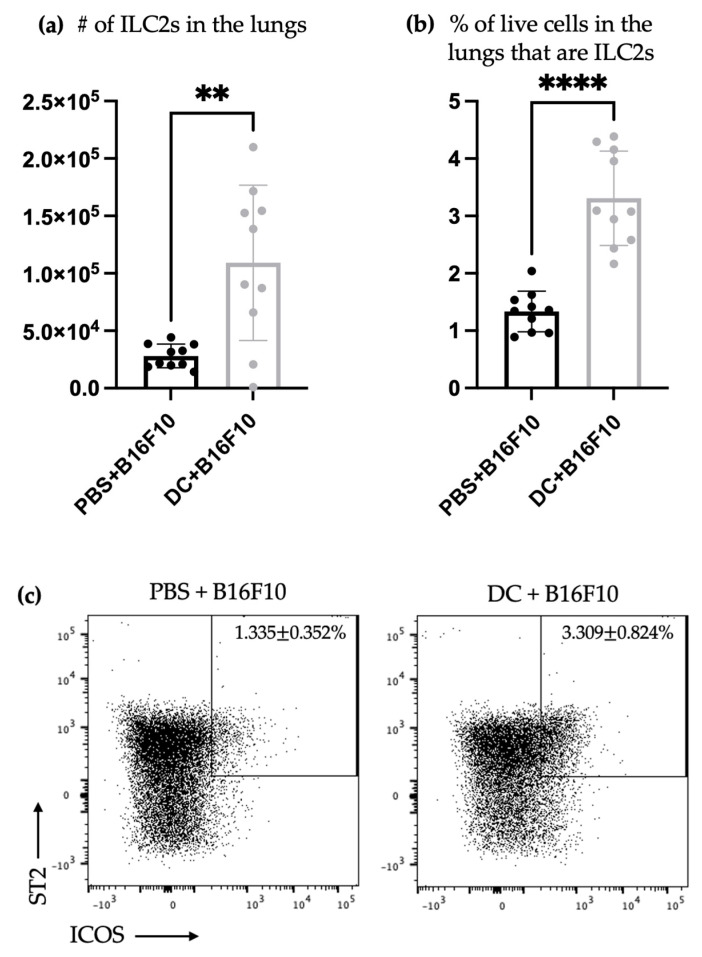
The number and proportion of ILC2s increased in the lungs after DC immunization, followed by a challenge with B16F10 melanoma cells. Female C57BL/6 mice were inoculated with DC vaccines injected into a hind footpad or were treated with phosphate-buffered saline (*n* = 10/group), and one week later, B16F10 cells were administered via injection into the tail vein. Lungs were harvested three days after challenge and analyzed for potential accumulation of ILC2s. The (**a**) number and (**b**) proportion of pulmonary ILC2s were enumerated. The data were analyzed using Student’s *t*-test; ** *p*-value < 0.005; and **** *p*-value < 0.0001. (**c**) Accumulation of CD45^+^Lineage^-^CD90.2^+^ST2^+^ICOS^+^ ILC2s in the lungs was monitored via flow cytometry (Appendix A). Representative dot plots showing the percentage of ILC2s. Lungs were also analyzed for ILC2 production of IL-5 and IL-13. Shown are the number of ILC2s producing (**d**) IL-13, (**e**) IL-5, or (**f**) IL-5 and IL-13. The data were analyzed using A two-way ANOVA test; *** *p*-value < 0.0005; and **** *p*-value < 0.0001. (**g**) Accumulation of CD45^+^Lineage^-^CD90.2^+^ST2^+^ICOS^+^ ILC2s producing cytokines in the lungs was monitored via flow cytometry. Representative dot plots show the percentage of all live cells in the lungs that were ILC2s. The ILC2s were assessed for production of IL-13^+^ and/or IL-5^+^. The error bars displayed in the graphs represent the standard deviation.

## Data Availability

Not applicable.

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
