# Peer review of "Dendritic Cell Vaccines Impact the Type 2 Innate Lymphoid Cell Population and Their Cytokine Generation in Mice"

_vaccines, 2023, doi:10.3390/vaccines11101559_

Round 1

Reviewer 1 Report

The work by Chang et al. addresses an interesting topic and the proposed results are potentially valuable. However, the experimental design and results shown are not always clear and, in some cases, incomplete. The discussion rewrites the results and misses important aspects that would add value to the results. I hope that the following suggestions will help the authors to improve the quality of their article.

-The analysis of minority populations such as ILC2 is complex and often requires enrichment strategies prior to Flow cytometry analysis. However, the authors did not consider this possibility and approached the quantification of the ILC2 population directly, using a cocktail of antibodies to exclude lineage+ cells, combined with the expression of 5 cell markers. The gates used for the analysis are included in the supplementary material. However, the percentages of each of the gates are not shown and in FigS1a, the selected lineage- population seems to represent 40%, which is not to be expected. In addition, the criteria for defining this gate appear to be very different in the different experiments shown in sup material. Although it is indicated that CD127 expression has been used, this criterion is not shown. Furthermore, the dot plots shown are of control mice and it would be more relevant to show the comparison of control mice and those inoculated with DCs.

-The information in material and methods is disorganised and unclear. For example, section 2.10 should precede 2.4 and include the strategy used to identify the ILC2 population. Section 2.4 should come before 2.7. Although the antibody references are very useful, the fluorochromes used should be included. The analysis software is not included.

- Section 2.8 states that the animals were sacrificed 7 days after tumour cell injection. It is confusing that later authors refer to day 10 (line 362).

- Figures 1 and 2 should be joined and Figure legends should be more clear. The dot plots are unclear as they show a percentage but it´s not possible to know what it refers to - total population or gate lin-CD45+CD90.2+ ST2+CD127+?

- Figure 1 shows the study of 2 mice, but 4 points are depicted

- In Figure 3. The displayed results cannot be understood. There should be 4 groups: PBS+PBS, PBS+B16F10, DC+PBS, DC+B16F10. Then it would be possible to analyse the effect of the tumour cells with or without the pre-vaccination.

- Given the striking increase in the number of ILC2 after vaccination, it would be relevant to phenotypically characterise them in more detail, including their isolation to study their transcriptional profile, at least in popliteal LN cells. This would shed light on one of the points that is still unclear in the study, namely the functionality of these cells.

- There is no discussion or analysis of the possible mechanisms by which the vaccine used would increase the number of ILC2 at the systemic level. Nor are the vaccinated DCs characterised. Did the increased secretion of IL5 and IL13 by ILC2 have a systemic outcome? Did vaccine increase IL33 or IL25? Does ILC2 increase in the bone marrow?

Reviewer 2 Report

This paper describes some features of the biology of the ILC2 helper-like innate lymphoid cells following the administration of a Dendrtic Cell (DC)   vaccine in female C57BL/6 mice. In particular, it deals with  the changes in ILC2 populations and the cytokines they produce, namely IL-5 and IL-13.

The authors have already made previous related observations in this field.

The paper is well written and understandable, and presented with well designed figures. The methods are clearly described.

The observations made are supposed to be of help regarding the comprehension of the behaviour of DC vaccines in humans and for the optimization of cancer immunotherapies.

Author Response

This paper describes some features of the biology of the ILC2 helper-like innate lymphoid cells following the administration of a Dendritic Cell (DC)   vaccine in female C57BL/6 mice. In particular, it deals with  the changes in ILC2 populations and the cytokines they produce, namely IL-5 and IL-13.

The authors have already made previous related observations in this field.

The paper is well written and understandable, and presented with well designed figures. The methods are clearly described.

The observations made are supposed to be of help regarding the comprehension of the behaviour of DC vaccines in humans and for the optimization of cancer immunotherapies.

Response: We want to extend our appreciation to the meticulous reviewer for the comprehensive assessment of our manuscript and invaluable input.

Reviewer 3 Report

This paper sought to understand the impact of a DC vaccine on ILC2 populations and cytokine generation in mice. The study would benefit from increasing the breadth of ILC populations, cytokines studied and tumour models used. It has greater insight into the impact of DC on ILCs than improving DC vaccines in which scenario the research is limited and unlikely to inform clinical translation. My recommendation is due to the preliminary nature of the report and the need for in depth investigation of other factors (cytokines, ILC populations, tumour models) to really demonstrate the complex nature of the interplay between DC and ILC2.

The introduction gives an overview of the concepts used in the paper, namely ILCs and specifically ILC2s, DC, DC vaccination and the need to improve this line of immune therapy.

The research seeks to understand the influence of DC vaccination on ILC2 in naïve and tumour challenged mice. The arguments in the introduction provided a confused message for the question being addressed. It was not clear whether the literature was more supportive of DC recruitment of ILCs to the tumour or whether ILCs (already in the tumour) trigger DC migration to LN. Given DC vaccination is about a translational aspect of research, timing – when the DC vaccine is most effective – is critical. The references covered many of the needed aspects of these concepts.

The plasticity of ILC is a hot topic. There is literature quoted in the reference list suggesting ILC2 can differentiate into ILC1 and ILC3 cells in certain environments. This concept needs to be addressed in the paper to put the results in context. The case for only addressing ILC2s in the study needs to be strengthened. The work investigates selected parameters (i.e. IL-13 and IL-5) without a rationale for excluding other important cytokines.

Improving DC vaccines is a translational field of research and there needs to be a clear assessment of how the results will be used to improve real world outcomes. The research design uses a particular in vitro derived DC preparation that whilst common in mouse studies does not reflect the in vitro derived cells that are used in human DC vaccines.  The DC prep is heterogenous and will impact other cells differently to other types of DC preps (i.e. FLT3L derived).

DC vaccines for cancer have yet to be successful in the clinical environment. They are predicted to be important in a therapeutic capacity. This paper designed the experiments in a way that the DC vaccines were used prophylactically. This does not reflect a likely pathway and justification for this needs strengthening.

The lack of focus on other ILC populations and production of other cytokines limits the significance of the findings.

The methods were generally well described although some details were missing. 2.6 Line 162 – DC vaccines were prepared from what? Line 186 – the timeline needs to be clarified to indicate if the experiment (from the day DC were injected) is 14 days (interpreted from line 186) or 21 days (interpreted from line 189).

The results are generally present in a clear manner. The error bar type (SD or SEM) used on the graphs needs to be included in the legends or methods.

The timeline in figure 1c only goes to day 7 and it would be interesting to see what happens on days 14 and 21 given the tumour challenge in later figure includes these days.

The methods do not describe how the experiments were conducted in terms of cohort sizes. Were the results from one experiment or representative of 3 or more? Does the experimental design explain variability in the results using the DC vaccination?

How were the gates determined in figures 2c, 3c, 4c? What is the significance of the high and low MFI populations in the IL-5+, IL-13- gates in fig 3g? Line 321 (3g) The figure does not show the accumulation of the CD45+ … cells but shows the gating used. Can the scale on the axes in figure 4g be reformatted to the same as that in the previous figures?

Line 338 suggests that DC vaccines influences ILC2s by migration, recruitment or local proliferation. The work does not comment of the plasticity of ILCs and other populations that may be contributing by differentiation to the outcomes.

Minor editing to improve some sentence structure would improve clarity. Examples – Line 110-112 The supernatant…., line 369 – what exactly is an “undetected increase”.

Round 2

Reviewer 1 Report

Although the authors did not add any of the experiments suggested, the article is now of sufficient quality.